# Long Prehensile Protrusions Can Facilitate Cancer Cell Invasion through the Basement Membrane

**DOI:** 10.3390/cells12202474

**Published:** 2023-10-18

**Authors:** Shayan S. Nazari, Andrew D. Doyle, Christopher K. E. Bleck, Kenneth M. Yamada

**Affiliations:** 1Cell Biology Section, National Institute of Dental and Craniofacial Research, National Institutes of Health, Bethesda, MD 20892, USA; 2Electron Microscopy Core, National Heart, Lung, and Blood Institute, National Institutes of Health, Bethesda, MD 20892, USA

**Keywords:** basement membrane, cell protrusion, invasion, cytoskeleton, contractility, spheroid, collagen, 3D culture, integrin, myosin II

## Abstract

A basic process in cancer is the breaching of basement-membrane barriers to permit tissue invasion. Cancer cells can use proteases and physical mechanisms to produce initial holes in basement membranes, but how cells squeeze through this barrier into matrix environments is not well understood. We used a 3D invasion model consisting of cancer-cell spheroids encapsulated by a basement membrane and embedded in collagen to characterize the dynamic early steps in cancer-cell invasion across this barrier. We demonstrate that certain cancer cells extend exceptionally long (~30–100 μm) protrusions through basement membranes via actin and microtubule cytoskeletal function. These long protrusions use integrin adhesion and myosin II-based contractility to pull cells through the basement membrane for initial invasion. Concurrently, these long, organelle-rich protrusions pull surrounding collagen inward while propelling cancer cells outward through perforations in the basement-membrane barrier. These exceptionally long, contractile cellular protrusions can facilitate the breaching of the basement-membrane barrier as a first step in cancer metastasis.

## 1. Introduction

Cancer metastasis is the major cause of mortality in cancer patients [1]. Metastasis occurs through a multi-step process generally consisting of cancer-cell invasion into an extracellular matrix, the intravasation and extravasation of cells in and out of the bloodstream, respectively, and the growth of secondary tumors in distant organs [2,3,4]. The initial event in carcinoma-cell invasion involves the breaching of a basement membrane (BM). The BM is a dense sheet of specialized extracellular-matrix proteins that separate epithelial compartments from the surrounding collagenous matrix [5,6]. It is also a nano-porous structure, with pore sizes that can vary depending on the type of tissue [7,8]. In the mammary gland, BM pores can be ~10 nm [9]. Since the average width of a cell is closer to ~10 µm, invasion would require the extensive expansion of BM nanopores.

Cancer cells can chemically degrade the BM using matrix metalloproteinases (MMPs) to generate pores that are sufficiently wide for invasion [10]. However, clinical trials using protease inhibitors failed to prevent metastasis and mortality, suggesting that cells might invade using mechanisms other than proteolysis [11]. In fact, developmental models including C. elegans [12], salivary glands [13], and mouse embryos [7] have established that cells or their blebs can perforate basement membranes by mechanically remodeling the matrix. However, such embryonic cells do not fully traverse the basement membrane. The mechanisms used by cancer cells for initial translocation across BM barriers into the surrounding extracellular matrix are not well understood.

Other aspects of cancer-cell invasion have been studied extensively. Cells can leave the primary tumor either individually or collectively [4]. Studies of individual cancer cells attempting to breach the BM indicate that they often require assistance from other cells, such as cancer-associated fibroblasts [14] or macrophages [15], to breach the BM, whereas large aggregates of cancer cells can disrupt the BM using proteases and physical forces [8,16,17]. In vivo patterns of invasion have been characterized using intravital microscopy [18,19]. However, the use of in vivo microscopy to determine how carcinoma cells initially breach the basement membrane and begin to enter surrounding collagenous matrices has been difficult due to intrinsic technical problems with accessibility, the low frequency of breaching events, resolution, and difficulties in performing localized experimental tests to probe mechanisms with sets of chemical inhibitors.

An alternative approach is to use in vitro 3D human-tumor spheroid models [20]. Time-lapse live imaging with long-working-distance objectives permits the imaging of the initial cell dynamics of three-dimensional spheroids in collagen gels breaching the BM. Using this 3D spheroid model, we previously established that human cancer cells in a spheroid perforate the basement membrane and expand these perforations during cell invasion using proteolytic, mechanical, and cytoskeletal mechanisms [21].

Cancer-cell spheroids are known to exert forces on a surrounding three-dimensional polymer network to facilitate tumor invasion [22,23,24], although the specific mechanisms are not clear. One mechanism might involve individual cell contractility. We and others previously showed that single migratory cells can locally deform a 3D collagen matrix by applying local traction force to the fibrillar collagen [25,26,27,28,29]. Here, we show that human cancer spheroid cells can send out long, slender, actin-based protrusions stabilized by tubulin through the basement membrane. These protrusions adhere to collagen fibrils using the integrin α2β1 to mechanically pull on and displace inward (centripetally) the surrounding collagen matrix using myosin II contractility. This centripetal collagen displacement is accompanied by cell breaching through the BM and outward cell translocation into the adjacent 3D collagen. During this process, human carcinoma-cell spheroids display a transient corona of long, arm-like, contractile cell protrusions that permit the breaching of the BM and the direct translocation of individual or collectively invading cancer cells through the BM barrier.

## 2. Materials and Methods

### 2.1. Cell Culture and Media

The cell lines used included human breast carcinoma MDA-MB-231 bone (MDA-MB-231BO) cells originally described in [30] and obtained from Dr. Kandice Tanner, National Cancer Institute, as well as human oral squamous cell carcinoma SCC9 cells obtained from the Ashok Kulkarni laboratory, NIDCR, NIH. The cells were confirmed to be MDA-MB-231 and SCC-9 cells, respectively, by STR analysis (ATCC, Manassas, VA, USA).

Culture media consisted of Dulbecco’s MEM (DMEM; Gibco,, Grand Island, NY, USA) with 10% fetal bovine serum (Life Technologies, Frederick, MD, USA), 1% penicillin/streptomycin (Life Technologies), and 1% L-glutamine (Life Technologies). Media were sterile-filtered through 0.45 µm nitrocellulose filters. Cells and spheroids were maintained in a humidified 10% CO_2_ incubator at 37 °C. Cells were confirmed to be free of mycoplasma using MycoAlert^®^ Mycoplasma Detection kits from Lonza (Rockville, MD, USA).

### 2.2. Three-Dimensional Spheroid Cell Culture

Detailed protocols for this method were published previously [20]. Briefly, MDA-MB-231 BO cells were seeded at 500 cells per well in ultra-low-attachment V-bottom (or U-bottom) 96-well plates (PrimeSurface from S-BIO, distributed by Vaupell in Hudson, NH, USA). After 8 h at 37 °C, the plates were centrifuged for 5 min at 300 RPM (18× *g*) at room temperature and placed back into the tissue-culture incubator for 48 h to permit the cells to form a compact spheroid via cell–cell adhesion. We then added Matrigel diluted to a final concentration of 5% in medium per well and centrifuged for 5 min at 300 RPM. The plate was then incubated at 37 °C for at least another 48 h to allow the cells to assemble a basement membrane around the spheroid. The spheroids were then washed in cold HBSS (Hanks balanced salt solution, Life Technologies) at least 3 times and embedded in 3 or 4 mg/mL rat-tail collagen I gels in MatTek 20 mm or 12-well, 1.5-thickness glass-bottom dishes. The gels were first placed on ice for 30 min, and then moved to the incubator to polymerize at 37 °C for 1 h. After gel polymerization, the serum-containing cell media or imaging media were added to the dish after the incubation period. We also tested 2 mg/mL collagen gels [21], but we ultimately chose to use 3 or 4 mg/mL gels because they proved less likely to tear or detach from our MatTek culture dishes during the rigorous washing in our immunostaining protocol.

### 2.3. Inhibitors

For experiments in which spheroids were treated with inhibitors, following spheroid polymerization in collagen gels, media were added containing the following treatments: BB94 (5 µM), GM6001 (20 µM), TIMP2 (4 µg/mL), TIMP3 (4 µg/mL), Y-27632 (20 µM), blebbistatin (20 µM), ML141 (20 µM), cytochalasin D (2 µM), latrunculin A (200 nM), CK-666 (100 µM), Fascin-G2 (50 µM), and nocodazole (660 nM). Furthermore, DMSO was used as the vehicle control for the inhibitors that were solubilized in DMSO. Anti-functional integrin antibodies, including integrin α2β1 and integrin β1, were used as inhibitors of integrins; rat monoclonal anti-β1 antibody and mouse monoclonal anti-α2β1 antibody clone BHA2.1 were obtained from MilliporeSigma, Rockville, MD, USA. We needed to use small-molecule and antibody inhibitors rather than knockout or knockdown approaches to ensure we could apply each treatment acutely in order to avoid long-term cellular changes in gene expression, proliferation, and morphology.

### 2.4. Immunostaining

Spheroids embedded in collagen gels were fixed using 4% paraformaldehyde in Dulbecco’s PBS with calcium and magnesium for at least 1 h, and then washed with Dulbecco’s PBS and blocked with 10% donkey serum for at least 1 h. Primary antibody was added to the dish and incubated at 4 °C overnight. Mouse monoclonal anti-myosin X antibody (C-1), rabbit monoclonal recombinant anti-non-muscle myosin IIA antibody, and rabbit polyclonal anti-AIF antibody were purchased from Abcam. Rabbit monoclonal recombinant anti-non-muscle myosin IIB antibody was obtained from Cell Signaling. Goat polyclonal anti-collagen type IV antibody, mouse monoclonal anti-PDIA3 antibody, and mouse monoclonal anti-α-tubulin antibody (clone DM1α) were purchased from MilliporeSigma. After washing the 3D spheroids with PBS, the collagen-embedded spheroids were incubated with the appropriate secondary Fab antibodies (Jackson ImmunoResearch, Westgrove, PA, USA) for at least 4 h at room temperature before imaging.

### 2.5. Confocal Imaging

All immunofluorescence confocal imaging was performed on a Zeiss LSM 880 with Airyscan super-resolution-imaging capabilities. The system was controlled with ZEN software version 2.3. Protrusions were imaged using a 63×, 1.27 NA Plan Apo water-immersion objective. Laser lines of 405 nm, 488 nm, 561 nm, and 640 nm provided illumination for Alexa Fluor 488, Rhodamine Red X, and Alexa Fluor 647 fluorophores, respectively. We acquired Z slices every 0.5 µm over a total distance of 10 µm Z. Some images were tiled 4 × 4 or 2 × 2. All images shown are maximum-intensity projections and were processed using ImageJ/FIJI version 1.54f.

### 2.6. Live-Cell Microscopy

For all live-cell fluorescence-imaging experiments, FluoroBrite DMEM (Gibco) with 10% FBS was used and supplemented with a 1:100 ratio of Oxyfluor (Oxyrase, Mansfield, OH, USA) with 10 mM DL-lactate (Sigma-Aldrich, St. Louis, MO, USA) as a substrate to reduce photobleaching and phototoxicity. Cells were imaged with a spinning-disk confocal scan head (Yokogawa CSU-X1, Sugar Land, TX, USA) automated Nikon Ti2 (Melville, NY, USA) microscope frame using a 40X APO-Chromat silicone oil objective (N.A. 1.15) to reduce spherical aberration in 3D. A Lun laser launch provided 405, 442, 488, 514, 561, and 647 laser lines. The primary dichroics (442/514/647and 405/488/568/647) were from Semrock (Rochester, NY, USA). Images were captured using a Prime 95B back-thinned CMOS camera (Photometrics, Tucson, AZ, USA) in 16-bit mode. A motorized Z-piezo stage (PI [Physik Instrumente], Auburn, MA, USA) was used to rapidly capture Z-stacks every 2 μm over a Z-distance of 60 μm. An environmental chamber surrounding the microscope-maintained cells at a constant 37 °C, with 10% CO2 and approximately 50% humidity (Okolab, Sewickley, PA, USA). We used NIS elements to control all hardware.

### 2.7. Widefield Time-Lapse Imaging

The brightfield live-cell images were obtained using a Nikon Ti-E inverted microscope (Melville, NY, USA) with motorized stage (Prior, Rockland, MA, USA) using 10× (N.A. 0.3) and 20× (N.A. 0.75) air objectives. Images were acquired with a Hamamatsu Orca Flash 4.0 CMOS camera (Hamamatsu-city, Japan). We used NIS-Elements (Nikon, Melville, NY, USA) to control all equipment. A red filter (high-pass 600 nm) was used to block lower wavelengths of light during experiments using blebbistatin or Y-27632. An environmental chamber (Precision Plastics, Beltsville, MD, USA) maintained cells at constant levels of 37 °C, 50% humidity, and 10% CO_2_.

Particulate phase-dense aggregates present in our collagen preparations were found to be useful as fiduciary markers to use in parallel to visualizing the collagen-matrix fibers themselves. Because the fibers were so clear, the kymograph analyses were based on collagen-fiber translocation, and the fiduciary particles were used only as highly visible markers of collagen movement, as shown in the movie montage in Figure 1.

### 2.8. Two-Photon Severing of Cell Protrusions

For two-photon cell-severing experiments, a Nikon A1R HD MP system was used (Nikon Instruments, Melville, NY, USA). Imaging used a 40× (1.15 N.A.) water-immersion objective and 488 nm (0.5–1.5%) and 561 nm (1–2%) laser lines to illuminate TAGGFP2-LifeAct and Atto565-labeled collagen, respectively, using resonant mode and bidirectional scanning at 512 × 512. We used NIS-Elements (Nikon) to control all equipment. Prior to imaging, a line spot was created approximately 10–15 μm behind the protrusion edge. Using ND acquisition functions, three sequences were configured: (1) a pre-severing 3D single timepoint (0.5-μm Z spacing over 40 μm), (2) a single-plane cell-severing sequence imaged at 7.5 frames/s for two minutes (with a 10 s delay prior to severing), and (3) a post-severing 3D single timepoint (same as the first). A Coherent (Glasgow, UK) Chameleon Vision II two-photon laser was set to 800 nm and 80% power, and a single point was chosen and ablated for 2 s. Kymographs were created and distances were measured at 2 s and 60 s after the 2 s ablation.

### 2.9. Focused Ion-Bean-Scanning-Electron Microscopy (FIB-SEM)

To prepare the samples for FIB-SEM, we followed the same precise procedure as previously described [31], with only a few minor adjustments. After fixing the samples, we carefully detached the coverslips from the cell-culture dish. Subsequently, we removed the samples and hydrogel from the coverslips and cut them into smaller pieces for further processing. Next, we stained the samples in a reduced osmium solution and incubated them in thiocarbohydrazide and osmium tetroxide. We allowed the samples to rest in uranyl acetate overnight at a low temperature of 4 °C and then stained them in Walton’s lead aspartate. Subsequently, we proceeded to dehydrate the samples in a series of ethanol and propylene oxide before incubating them in Epon overnight and then for three hours in 100% Epon. Finally, we placed the samples on aluminum Zeiss SEM stubs for polymerization in an oven at 60 °C for two days. Once the samples were dry, we coated them with 40 nm gold and then imaged them using a Zeiss Crossbeam 540 FIB-SEM microscope. Platinum and carbon were deposited over the region of interest, and the run was set up and controlled by Atlas 5 software (Zeiss version 5.3.5.3 x64 via Fibics Incorporated, Ottawa, Canada). SEM settings: 1.5 keV; 2.0 nA; milling probe: 700 pA, 30 keV. The slice thickness and voxel size were set to 20 nm.

### 2.10. Collagen-Displacement Analysis

To quantify collagen displacement semi-automatically, a Fiji (ImageJ) macro was created by A.D.D. In brief, 4 regions of interest were selected automatically by drawing two lines from each corner to corner of the timelapse file. The kymograph-maker plugin was then used to create a kymograph of the region over time. This provided 6 regions for each spheroid selected in a similar manner for each file in every treatment and control group. These kymographs were then opened in a separate macro, where we measured the distance of change (displacement) in microns of the collagen from time 0 to 18 h, as indicated by the dashed lines similar to those in Figure 2C. We drew 4 lines from time 0 to 18 h on each kymograph and measured the distance of displacement. The lines were drawn from both sides of the spheroid and were picked at a similar region each time, close but not touching the spheroid. This process was semi-automated using a macro written by A.D.D.

### 2.11. Quantification of Invasion by Counting Nuclei

Spheroids were imaged for 18 h at the equator using live timelapse phase contrast and fluorescence microscopy. At the 18 h timepoint, we counted all nuclei stained with SiR-DNA within a 100 µm Z-range of 10 µm Z-optical slices at the equator.

### 2.12. Protrusion Quantification

Using phase timelapse videos at the 8 h timepoint. When protrusions appeared in controls, we measured and counted each protrusion around the entire perimeter of each spheroid. Each spheroid was imaged at the equator within a 100 µm zone using 10 µm Z-slices.

### 2.13. Statistical Analysis

We repeated each experiment independently at least 3 times (listed as N), and each experiment contained at least 3 spheroids (listed as n). Collagen displacement was measured from the average displacement of four independent sides of each spheroid. For the statistical analyses, we used one-way ANOVA with Dunnett’s post hoc test. Prism 4 by GraphPad Prism software (version 9.3.1, Boston, MA, USA) was used for all graphs and statistical analyses.

## 3. Results

### 3.1. Long Cellular Protrusions from Cancer Spheroids Traverse Basement Membranes into Collagen Matrix, and They Pull the Matrix Centripetally

We report that when utilizing cancer cells in a spheroid encapsulated by a basement membrane and surrounded by a 3D collagen matrix to model initial tumor invasion [20], the cells initially send out long protrusions up to 100 µm long that are highly visible by phase contrast microscopy 10–12 h after embedding in a 3–4 mg/mL collagen hydrogel (Figure 1A,B). These protrusions eventually form a corona of slender cell protrusions around the spheroids, and they increase in number up to 24–32 h, accompanied by a global inward movement of the surrounding collagen gel (Figure 1A,B and Appendix A). Timelapse microscopy (Appendix A) revealed that as the numerous cellular protrusions extend through the basement membrane and into the surrounding collagen, fiduciary inclusions (black arrowhead in Figure 1A) are translocated inward toward the spheroid basement membrane. These findings suggest that cells might use these long protrusions to generate tension/stress against the collagen, displacing it inward as they migrate outward during matrix invasion. It should be noted that the protrusions are dynamic (Appendix A), and they extend and retract at an average rate of 21–25 µm per hour (Appendix A). Moreover, the edge of the spheroid also expands during invasion (Appendix A).

The 3D confocal microscopy confirmed that the extensions of these tumor spheroids are unusually long, slender cellular protrusions (~2–4 µm in diameter and 30–100 µm long) that penetrate through the basement membrane (as depicted by the collagen IV staining) into the collagen gel prior to cell invasion (Figure 1C, left panel). As the cancer cells’ nuclei and bodies begin to squeeze through and traverse the basement membrane, these long protrusions are maintained (Figure 1C, middle panels; note that the nucleus becomes elongated), and they persist after the nuclei of the cells have traversed the basement membrane (Figure 1C, right panel). These findings suggest the hypothesis that the long protrusions pull the adjacent collagen matrix inward, toward the spheroid, to facilitate outward cell translocation, i.e., for the breaching of and squeezing through the holes in the basement membrane by the nucleus (the largest cellular organelle) and cell body for successful cellular invasion outward into the 3D collagen matrix.

### 3.2. Cells Pull on and Translocate the Surrounding Collagen Matrix toward the Basement Membrane Using Myosin II

To quantify the direction and extent of the inward, centripetal translocation of the collagen matrix surrounding the spheroids, we imaged the cancer spheroids with basement membranes using phase-contrast microscopy for up to 60 h. The kymograph analyses performed along the track of cells migrating out of tumor spheroids (Figure 2A,B) confirmed that the collagen matrix is actively translocated toward the spheroid over time. We established that within the first ~20 h, during which many protrusions reach through the BM into the matrix and are followed by cells moving outward into the collagen gel (as shown in Figure 1A,B), the collagen gel is concurrently pulled inward (Figure 2B, region 1). Eventually, widespread streams of collectively migrating cells leave the expanding spheroid and invade into the collagen gel (Figure 2B, region 2), while the collagen’s inward displacement continues.

To establish whether this inward collagen translocation toward cancer spheroids results from a mechanical contractile process, we treated the cells with blebbistatin, a potent and selective myosin II ATPase inhibitor, and imaged the spheroids for 18 h. The kymographs from the 18-h timelapse videos (Figure 2C–E) confirm that the inhibition of myosin II contractility with blebbistatin led to a substantial reduction in collagen displacement (90%) compared to spheroids treated with the vehicle control (Figure 2C–E). We found that the untreated cancer cell spheroids displaced the collagen centripetally at an average rate of ~20 µm per 18 h, whereas blebbistatin strongly suppressed the collagen displacement (~2 µm per 18 h; Figure 2E). In addition, the blebbistatin treatment resulted in the strong concomitant inhibition of outward cancer-cell invasion compared to the control (91%; Figure 2F and Appendix A). In other words, the loss of collagen displacement in blebbistatin-treated spheroids is linked to the suppression of cell invasion, suggesting that myosin II contractility is required for cancer cells to physically pull their bodies through the BM and into the surrounding collagen matrix during invasion by contracting their long protrusions attached to collagen fibrils.

We next tested whether this contractility-dependent phenomenon documented in a breast-cancer-cell line could be observed in a different type of cancer to evaluate the generality of this phenomenon. We found that SCC9 human oral carcinoma cells also generate long protrusions, collagen displacement/translocation of nearly 20 µm over a 36-h imaging period, and outward cell invasion into the collagen matrix. Treatment with blebbistatin strongly inhibits SCC9-mediated collagen displacement and cancer-cell invasion (Figure 2G,H).

To test whether the inward movement of collagen is mediated by individual cells generating this centripetal collagen displacement, we conducted experiments to sever individual protrusions (Figure 3A,B). The cells were visualized by transduction with lentiviral mNeon Green Lifeact, and spheroids with basement membranes were generated. Using two-photon confocal live-cell microscopy, we ablated single protrusions near the distal tip, which were associated with collagen fibrils and imaged rapidly every ~150 ms for several minutes. The kymograph analyses demonstrate that after ablation, the matrix undergoes relaxation, as shown by the collagen’s movement away from the spheroid (Figure 3C). We observed rapid, reproducible ~1-μm collagen gel displacements, in which the matrix snapped back away from the spheroids within 2 s after the laser ablation of the protrusion, consistent with a loss of localized tension (Figure 3D and Appendix A).

### 3.3. Actin Polymerization and Tubulin Are Necessary for Long-Protrusion Collagen Displacement and Cell Invasion

The confocal microscopic examination of the F-actin localization established that the long, slender protrusions are enriched in actin (Figure 1C). To test the involvement of actin in the protrusion formation and collagen displacement, we directly inhibited the actin polymerization using two actin inhibitors, latrunculin A and cytochalasin D. The treatments with each inhibitor resulted in the disappearance of the protrusions and a dramatic inhibition of the collagen displacement (Figure 4A,C,D). Other cytoskeletal inhibitors targeting actin-associated proteins, such as Cdc42, fascin, and Arp2/3, also significantly decreased the collagen displacement (Figure 4A,B) and invasion (Figure 4D) compared to the controls. The inhibition of ROCK (Rho-associated protein kinase), which plays major roles in regulating myosin II contractility [32,33], but can also affect actin polymerization [34], also substantially inhibited the inward collagen displacement (Figure 4B). The quantification 8 h after implanting the spheroids in a collagen matrix demonstrated that blebbistatin, latrunculin A, and nocodazole all suppress cellular protrusions by more than four-fold. After 8 h of timelapse live imaging, when protrusions through the BM were present prior to the nuclear translocation, we quantified the numbers of protrusions more 30 µm long (Figure 4C,D). For all the spheroids, we counted the number of long protrusions within a 100-μm Z-dimension zone at the equators of the spheroids imaged by phase microscopy. In Figure 4C, we show representative images of parts of the spheroids for comparison. We found that the myosin II inhibition substantially decreased the number of long protrusions by approximately 81%, and that the inhibition of the cell cytoskeletal components actin and tubulin blocked the extension of long protrusions completely from the spheroids (Figure 4D).

We then investigated whether these long protrusions were stabilized by microtubules during their formation, e.g., analogously to the stabilization of axons and neurites. The treatment of spheroids with the tubulin disruptor nocodazole inhibits the formation of long protrusions, as well as inhibiting collagen displacement (Figure 4A,B). In a complementary experiment to test the role of microtubules in the stabilization of the protrusions, we permitted the spheroids to form long protrusions and then treated them with nocodazole to test for destabilization. After treatment for one day, the long protrusions shortened, and they disappeared after three days of treatment (Appendix A). Together, these data suggest that while an intact, contractile actin cytoskeleton is required for protrusion formation and/or BM breaching, microtubules may be necessary for protrusion stabilization.

### 3.4. Long Protrusions Require the Integrin α2β1 Collagen Receptor to Pull on the Matrix

To determine whether integrins were required to grip collagen fibrils for matrix displacement and cell invasion, we tested anti-functional integrin antibodies that are known to disrupt cell-to-collagen adhesion. The inhibition of an integrin subunit shared broadly with multiple integrin α subunits, integrin β1, substantially decreases the number of protrusions, the level of collagen displacement, and cell invasion (Figure 5A,B). More specifically, the inhibition of the collagen-binding integrin α2β1 to inhibit protrusion attachment to collagen also strongly inhibits collagen displacement and cell invasion (Figure 5A,B). These findings indicate that cancer spheroid cells attach to collagen via α2β1 to apply pulling forces during invasion.

### 3.5. Proteases and Contractility Have Complementary Roles in Protrusion Formation, Collagen Displacement, and Cell Invasion

We previously established that the inhibition of proteases, particularly MMPs, suppressed the formation and enlargement of perforations in the basement membrane, which blocked the translocation of cancer cells through the basement membrane [21]. However, it was not clear whether the long protrusions described here could still form and protrude through the tiny basement-membrane perforations of protease-inhibited spheroids, and whether they could still displace collagen using myosin II contractility. A quantitative comparison of the effects of multiple protease inhibitors revealed the abundant presence of protrusions and substantial collagen displacement that is only modestly decreased after the suppression of proteolysis, but with low levels of cell invasion (Figure 5C,D and Appendix A).

As noted previously concerning the role of contractility, cells retain some protrusions but fail to pull on and translocate the collagen matrix after inhibition with small-molecule inhibitors of myosin II (blebbistatin) and ROCK (Y-27632); the cells concurrently fail to translocate their cell bodies across the BM to invade into the surrounding collagen. The combination of the inhibition of proteases and myosin II contractility suppress both collagen displacement and cancer-cell invasion into the collagen gel slightly more effectively, suggesting that both contribute (Figure 5C,D and Appendix A).

### 3.6. In-Depth Characterization Reveals That the Long, Contractile Protrusions Are Packed with Organelles

We next focused on characterizing further the long, thin cellular protrusions extending from the cancer-cell spheroids through the basement membranes to be able to compare them with previously published cellular extensions. They vary in length and diameter, e.g., they are often 30–100 µm in length and ~2–4 µm in diameter. According to time-lapse microscopy, they can protrude at an average rate of about 25 µm ± 15 (SD) µm/hour, but they can also retract, followed by the protrusion of new cellular extensions (Appendix A, and Appendix A). Some protrusions may also have lateral blebs and bifurcations. The 3D-electron-microscopy reconstructions of these cellular protrusions revealed numerous subcellular structures not reported in previously described protrusions—instead, the protrusions represent long, slender, invasive protrusions or arm-like extensions of the cell body rich in subcellular organelles. We observed vesicles of varying sizes, mitochondria, and ER (endoplasmic reticulum) in the long protrusions (Figure 6A,B).

We confirmed by immunofluorescence confocal microscopy that mitochondria and ER were present in the long protrusions, with no obvious concentration at either end. Microtubule tubulin was observed along the entirety of each protrusion (Figure 6B), consistent with our inhibitor studies, which indicated that microtubules act to support these lengthy structures. Our inhibition studies also showed that myosin II is important for collagen displacement (Figure 2D,E), and we confirmed robust immunofluorescence staining for both myosin IIA and IIB in the protrusions (Figure 6B). Interestingly, we did not detect in these protrusions the myosin X marker for filopodia (Appendix A). Therefore, these long, slender protrusions are not related to filopodia, consistent with their content of mitochondria, ER, and vesicles. These long, thin cellular extensions instead appear to be slender, arm-like, force-generating extensions of the cytoplasm packed with contractile and supportive elements that can use cell-surface integrins to grip and pull on fibrillar collagen.

## 4. Discussion

In this study, we describe a mechanism that cancer cells in spheroids can use to squeeze through perforations in a confining basement membrane and invade using an unexpected strategy based on transient, exceptionally long, slender, and contractile cell protrusions. These cellular extensions are 30–100 µm long and can apply mechanical force to pull on the surrounding collagen matrix as cells traverse the basement membrane. In this manner, they displace the matrix centripetally toward the spheroid. These long, slender actin-based protrusions through the basement membrane are stabilized by microtubules and attach to the fibrillar collagen matrix via integrin α2β1. They require myosin II contractility to translocate the cell body through the basement membrane for outward matrix invasion. These long grasping/prehensile protrusions are relatively transient structures associated with very early invasion, since previous studies and our own have shown that these cells lack these long protrusions when migrating/invading as individual cells in a 3D matrix [25,35,36].

The cell protrusions that we describe are unusually long. They extend out from the spheroid in a dynamic extend-and-retract fashion at average rates of 25 µm per hour until they reach their final length of up to 100 µm. Published studies using a variety of cell lines, including MDA-MB-231 cells, have described other cellular protrusions that are shorter, often lack most organelles, and are associated with cell migration or communication. These other types of protrusions include microspikes [37], tunneling nanotubes [38], microtentacles [39], endothelial sprout tips [40], and other invasive protrusions ranging from 50 nm to 41 µm in length [23,37,38,39,40,41,42,43,44,45,46,47]. In a manner similar to our longer dynamic protrusions, some researchers report that their protrusions can retract and extend into the matrix [39,40]. Actin-based microspikes express invadopodial markers, such as cortactin and Tks5 [37], while other protrusions, such as microtentacles, contain F-actin and tubulin, but no myosin-X, distinguishing them from filopodia [39]; our longer protrusions also lack myosin-X. Interestingly, microtentacles are reported to lengthen during cytochalasin D treatment but are inhibited by nocodazole [39], which differs from the long protrusions we describe, which are instead inhibited by cytochalasin D, as well as by latrunculin and nocodazole.

In addition to their unusual lengths, the long, slender cancer-cell protrusions we describe in this study are actively contractile. Their disruption by a variety of cytoskeletal inhibitors, including cytochalasin, latrunculin, blebbistatin, and nocodazole, results in the inhibition of inward, centripetal collagen displacement. Blebbistatin produces the dramatic inhibition of collagen displacement and invasion in both MDA-MB-231BO and oral cancer SCC9 cell lines, according to quantitative kymograph analyses. These initial findings suggested that the cells use the protrusions to mechanically pull the matrix toward them as they begin invasion. In support of this concept, a two-photon laser ablation of the protrusions resulted in a reproducible outward 1-μm relaxation of the matrix within a minute, as expected for protrusion-mediated radial tension on the collagen.

Our identification of the key role of contractile cellular protrusions provides a mechanism with which to explain the strong inward collagen-matrix remodeling reported in previous studies using cancer spheroids [22,23,24]. In fact, potentially related protrusive activity can be seen in Video 2 in [24], suggesting that even though single cancer cells generally lack these long protrusions [25,35,36], their organization into a spheroid with or without a basement membrane can trigger their formation and function. This centripetal movement of collagen may also help to explain the original findings of Keely et al., who identified local collagen accumulations encircling tumors in vivo, which are classified as TACS1 [48], e.g., as cancers transition from in situ to invasive.

A third notable and unusual characteristic of these cancer-cell protrusions is that they are filled with a wide variety of intracellular organelles, as well as cytoskeletal proteins. Thus, they are long, arm-like extensions of the cell body rather than other types of protrusion that exclude cellular organelles, such as filopodia, lamellipodia, tunneling nanotubes, and microtentacles. The immunofluorescence staining of the long protrusions confirmed the presence not only of F-actin and α-tubulin, but also of mitochondria, endoplasmic reticulum, and vesicles. In addition, myosin IIA and myosin IIB are prominently identifiable by immunofluorescence, whereas myosin X is not found. Consequently, these long, prehensile cancer-cell protrusions appear to function as arm-like extensions of the cell body itself that can both extend and actively contract to pull on the extracellular matrix, unlike previously characterized thin anterior protrusive structures devoid of organelles, such as mitochondria.

How do these protrusions attach to collagen fibrils to permit contraction? We tested for the roles of integrins using anti-functional antibodies against integrins known to be receptors for collagen. The inhibition of integrin α2β1 effectively inhibited the formation of long protrusions. Collagen displacement and cell invasion are both dramatically concomitantly inhibited by α2β1 inhibition. These results indicate that integrin α2β1 is necessary for protrusion attachment and the collagen displacement associated with cancer-cell invasion into the collagen matrix.

In a previous publication, we used this cancer-spheroid model to characterize how cells perforate and expand perforations or holes in the basement membrane during cancer-cell invasion [21]. Protease inhibition suppressed perforation expansion, yet our current study reveals that collagen is still pulled centripetally inward. Although the basement-membrane perforations are smaller than the diameter of a cell, thus preventing invasion, the long slender protrusions are still present and functional.

We propose a unified model for this mechanism for the cancer-cell breaching of the basement membrane based on the literature and our current findings, which are summarized schematically in Figure 7 as follows: (1) Cells initially poke small holes in the basement membrane using their short actin-based protrusions and proteases, i.e., via invadopodia [49,50]. (2) The actin-rich protrusions then lengthen dynamically while stabilized by microtubules to extend through small perforations in the basement membrane and into the surrounding collagen matrix. (3) These protrusive extensions then attach to collagen fibrils via integrin α2β1, while the cell bodies are still confined within the spheroid (because the initial tiny holes in the basement membrane have diameters that are smaller than that of the nucleus, the largest organelle of the cell). (4) As further proteolysis enlarges the basement-membrane perforations, the cells can finally squeeze through the hole using myosin II to translocate their bodies through the basement membrane for invasion when using this long-protrusion contraction-based mode of cancer-cell invasion; nuclei squeezing through even narrow passageways can deform substantially without rupture [51]. (5) Once cells have traversed the basement membrane, they can undergo a variety of modes of invasion during single or collective cell migration and tumor expansion [24,52,53,54,55]. They can use a variety of other leading-edge protrusions, depending on the type of cancer, including filopodia, microtentacles, and blebs [38,39,52,53,55,56,57]. Their migration can be enhanced by other cell types [14,43,58,59], and they can adapt to differing collagen-matrix stiffnesses by altering their metabolism [60].

A limitation of this study is that it is based on only two cancer-cell lines. It will be necessary to extend these findings in the future to determine is the extent of the spread of this mechanism of protrusion-based BM breaching. Interestingly, because this mechanism involves only cancer cells, it avoids the need for helper cells, such as macrophages or cancer-associated fibroblasts, which was reported as a requirement for successful invasion across BMs when applying models using single, isolated cancer cells not present in a tumor/spheroid mass [14,15].

## 5. Conclusions

Our study established the concept that cancer cells in spheroids encapsulated by a basement membrane and embedded in collagen gels use long actin-based, prehensile protrusions that extend through basement-membrane proteolytic perforations to extend into the surrounding matrix. They lengthen these protrusions in a process requiring actin polymerization and stabilizing microtubules and then attach them to the collagen using integrin α2β1. These protrusions, once attached, can then apply an integrin-dependent pulling force to the matrix using myosin II contractility. The result is both inward collagen displacement and outward force on cells through the basement membrane holes, as the cells use myosin II and actin polymerization to translocate their bulky nuclei and the rest of their bodies outward into the matrix during invasion.

## Figures and Tables

**Figure 1 cells-12-02474-f001:**
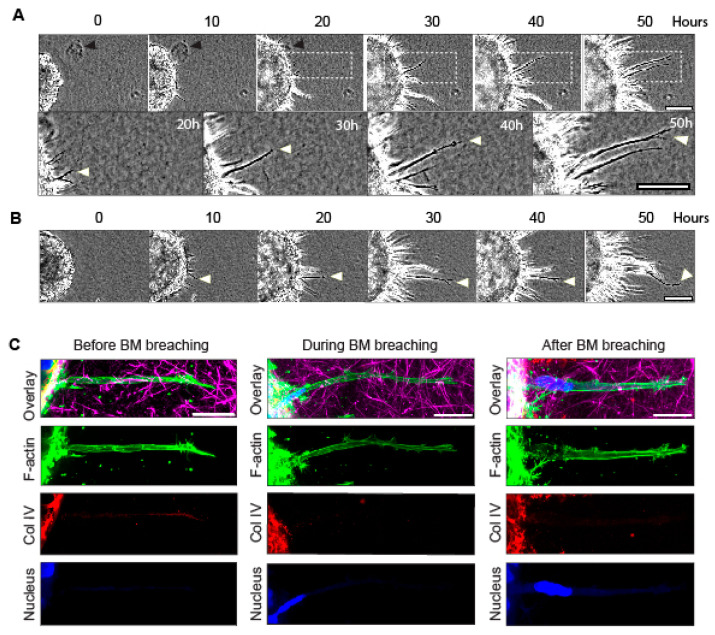
Timelapse imaging of spheroid-invasion assay and confocal imaging of long protrusions during initial breaching of the basement membrane. (**A**,**B**) Timelapse phase-contrast imaging of cancer-cell protrusions and invading cells extending from tumor spheroids over 50 h. Note that protrusions form between 0 h and 10 h. White arrowheads indicate a long protrusion of interest that extends over 30 h into the surrounding collagen gel, as the collagen is concurrently pulled toward the spheroid. Black arrowhead marks a fiduciary particle in the collagen that is translocated toward the spheroid during invasion. (**C**) Confocal immunofluorescence microscopy showing F-actin (green, stained with phalloidin), collagen IV (Coll IV, red), collagen I (magenta), and DAPI-stained nuclei (blue) of long protrusions generated by single-spheroid cancer cells that extend out of the basement membrane and into collagen gel before, during, and after cell breaching of the basement membrane (note the locations of the blue nuclei). Scale bars: (**A**), 100 µm; (**A** inset and **C**), 20 µm.

**Figure 2 cells-12-02474-f002:**
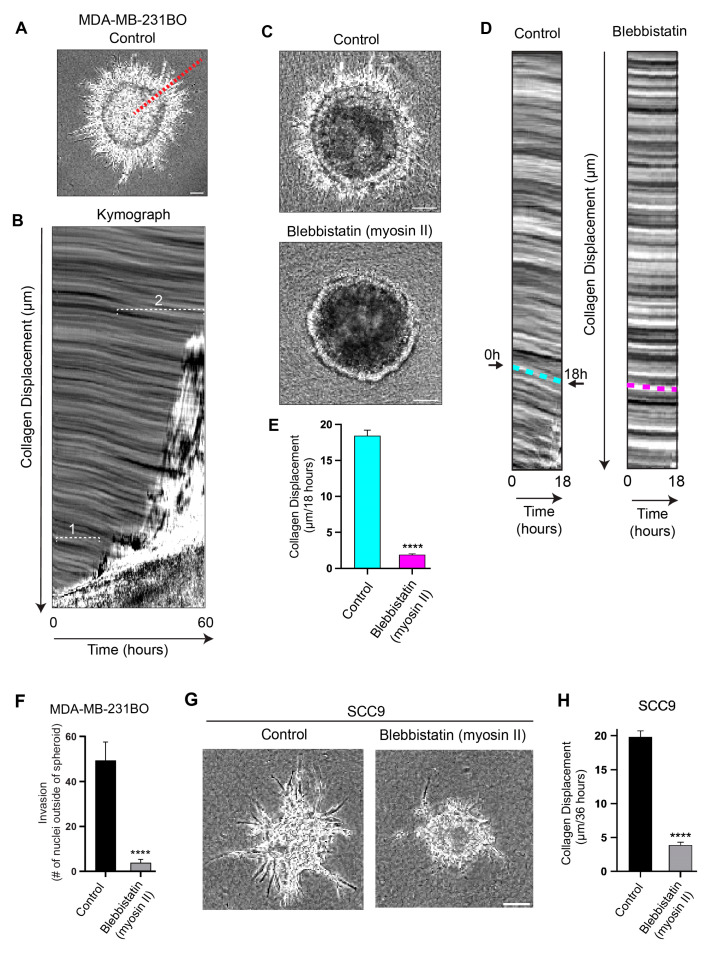
Collagen displacement and subsequent cell invasion require myosin II contractility. (**A**) A region of interest (red dashed line) used to generate a kymograph from a timelapse video of MDA-MB-231BO tumor spheroid cells invading through a basement membrane and into the surrounding collagen gel over 60 h. (**B**) Kymograph of spheroid in A demonstrating the movement of collagen over time: the *x*-axis = time; *y*-axis = distance in µm. Collagen movement towards the expanding spheroid (bottom of kymograph) occurs during early protrusion and subsequent cell invasion through the BM during the initial 18 h; collagen displacement (bracketed regions 1 and 2), is followed by outward streaming cells into the collagen gel (region 2). (**C**) Spheroids treated with the myosin II ATPase inhibitor blebbistatin or DMSO vehicle control imaged over 18 h. (**D**) Kymographs used to quantify collagen displacement in panel C. Cyan and magenta dashed lines indicate the extent of collagen movement over 18 h for control and blebbistatin-treated spheroids, respectively. (**E**) Kymograph analyses indicate collagen displacement was dramatically inhibited by the myosin II inhibitor compared to its control. (**F**) Quantification of numbers of nuclei invading outside the perimeters of the spheroids. (**G**) Comparisons of SCC9 oral cancer cells’ invasion into the collagen hydrogel in vehicle control versus blebbistatin over 36 h. (**H**) Kymograph analysis of SCC9-mediated collagen displacement comparing vehicle control to blebbistatin treatment. Collagen displacement was measured from four computer-selected regions surrounding each individual spheroid. Number of independent experiments N = 3; number of spheroids in each experiment n = 3–8. **** *p* < 0.0001. Mean value ± SEM. Scale bars: (**C**) and (**G**), 100 µm.

**Figure 3 cells-12-02474-f003:**
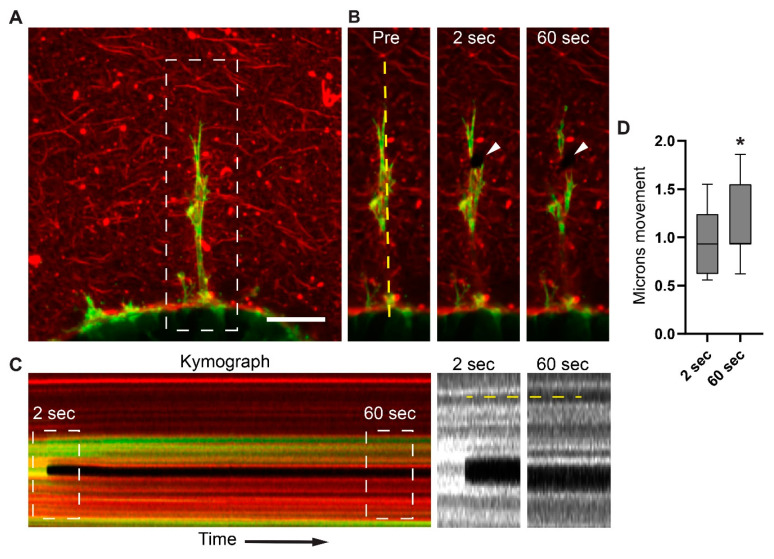
Long protrusions generate tension within the collagen microenvironment. (**A**) Maximum-intensity project (MIP) of MDA-MB-231BO cells expressing mNeon Green LifeAct (green). Eight hours after embedding spheroids in collagen gels, cells form protrusions into the surrounding collagen matrix (red). The image is a maximum-intensity projection of 10 µm. (**B**) The same cell (white dashed box in panel A) shown only at a single Z plane immediately before (Pre) and at 2 s and 60 s after a focused two-photon beam severs the protrusions (arrowheads). Vertical yellow dashed line indicates position of the kymograph shown in panel C. (**C**) Kymograph depicts the time of protrusion severing (black stripe) and the relatively small release of tension within the ECM after protrusion collapse. White dashed boxes (magnified on the right) illustrate the subtle changes 2 s and 60 s after cell and ECM severing. Vertical dashed yellow line demonstrates a small change at 2 s, followed by continued ECM relaxation over 60 s. (**D**) Analysis of collagen movement 2 s and 60 s after severing of a protrusion; box indicates lower and upper quartile and whiskers indicate 10th and 90th percentiles. N = 9, n = 48. * *p* ≤ 0.001. Scale bars: (**A**) and (**B**) 25 µm.

**Figure 4 cells-12-02474-f004:**
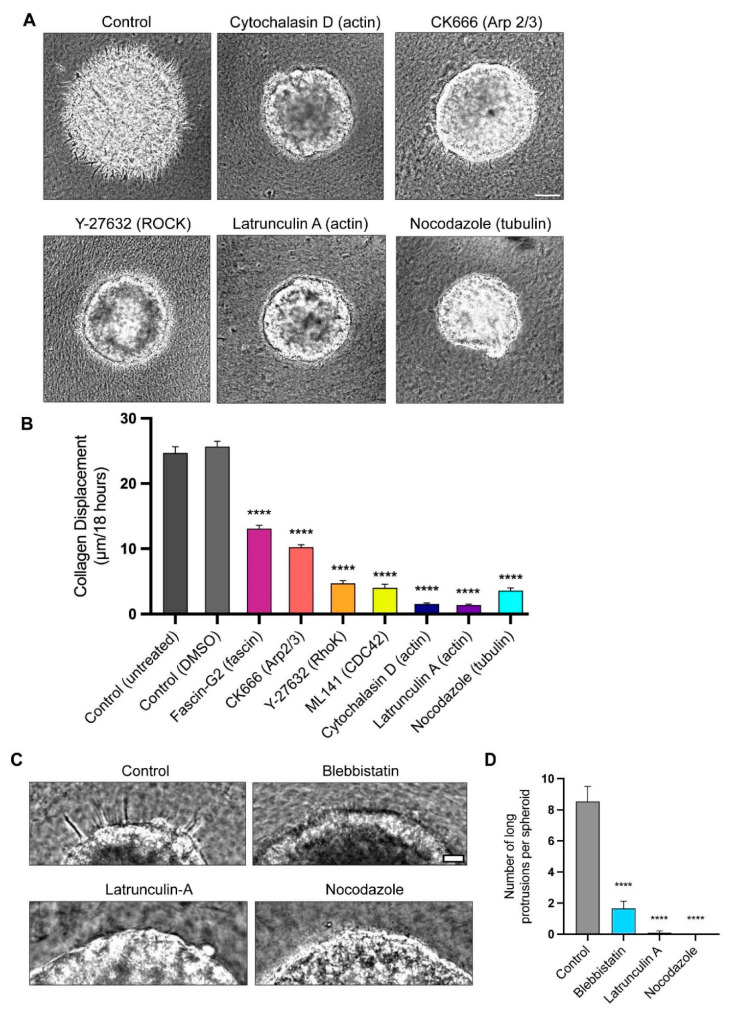
Quantitative comparison of effects of cytoskeletal inhibitors on rates of collagen displacement and protrusion count. (**A**) Representative images of spheroids 18 h after treatment with different inhibitors that directly or indirectly affect cytoskeletal components. (**B**) Kymographs were generated for each condition, and the distance of collagen displacement from time 0 to 18 h was quantified. (**C**) After 8 h of timelapse live imaging, control spheroids extended protrusions into the surrounding collagen. However, spheroids treated with blebbistatin, latrunculin A, and nocodazole failed to generate extensive protrusions. (**D**) Quantification of long protrusions shown in panel C, defined as 30 µm or longer, extending from each spheroid after control and inhibitor treatments. Data are based on pooled data from 3 independent experiments with at least 3 spheroids per experiment; similar results were obtained in all 3 experiments. Statistical analysis by one-way ANOVA with Dunnett’s test in comparison to the DMSO control: **** *p* < 0.0001, N = 3, n = 3–7. Mean value ± SEM. Scale bars: (**A**) 100 µm, (**C**) 30 µm.

**Figure 5 cells-12-02474-f005:**
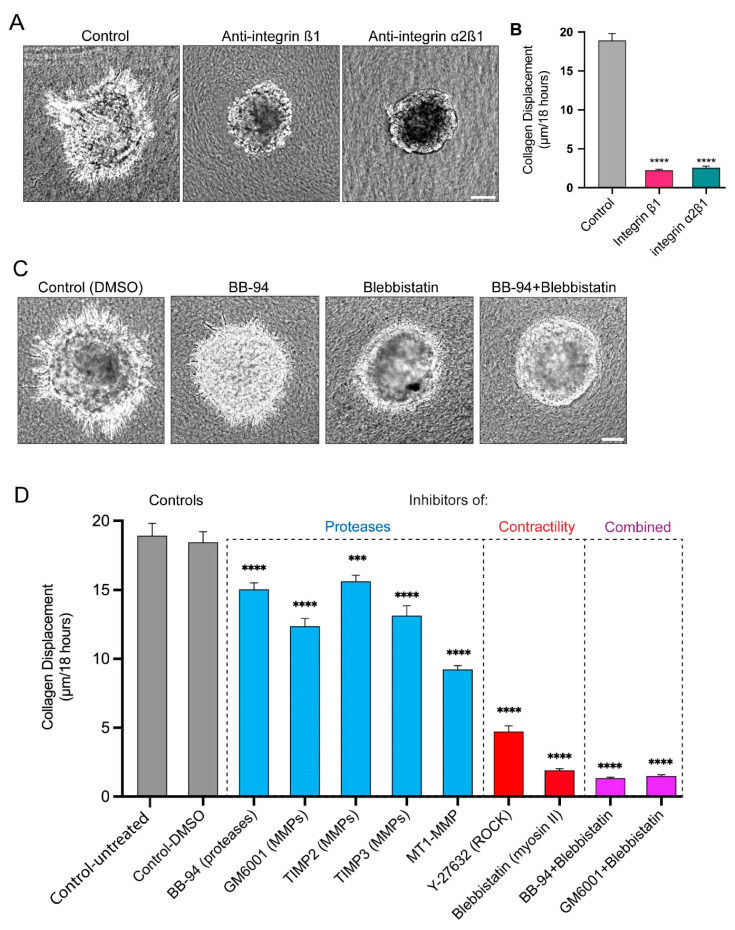
Key role of attachment via α2β1 integrin in collagen displacement, and comparison of contributions of proteases versus myosin contractility in collagen displacement and invasion. (**A**) Representative images of inhibition by integrin β1 and α2β1 anti-functional antibodies of spheroid invasion into collagen gels compared to controls. Spheroids were treated with function-blocking antibodies to integrins and then imaged over 18 h. (**B**) Collagen translocation from time 0 to 18 h for each condition. (**C**) Representative images of spheroids after 18 h treatment with a broad protease inhibitor (BB-94), and/or myosin inhibitor (blebbistatin) and their effects on cell protrusions and invasion. (**D**) Collagen displacement after treatment with broad-spectrum protease inhibitors, tissue inhibitors of metalloproteinases (TIMPs 2 and 3), contractility inhibitors of myosin II and ROCK, and combinations of BB-94 and GM6001 with blebbistatin. Statistical analysis by one-way ANOVA with Dunnett’s test in comparison to the DMSO control: **** *p* < 0.0001, *** *p* < 0.0008, N = 3, n = 3–7. Mean value ± SEM. Scale bar: 100 µm.

**Figure 6 cells-12-02474-f006:**
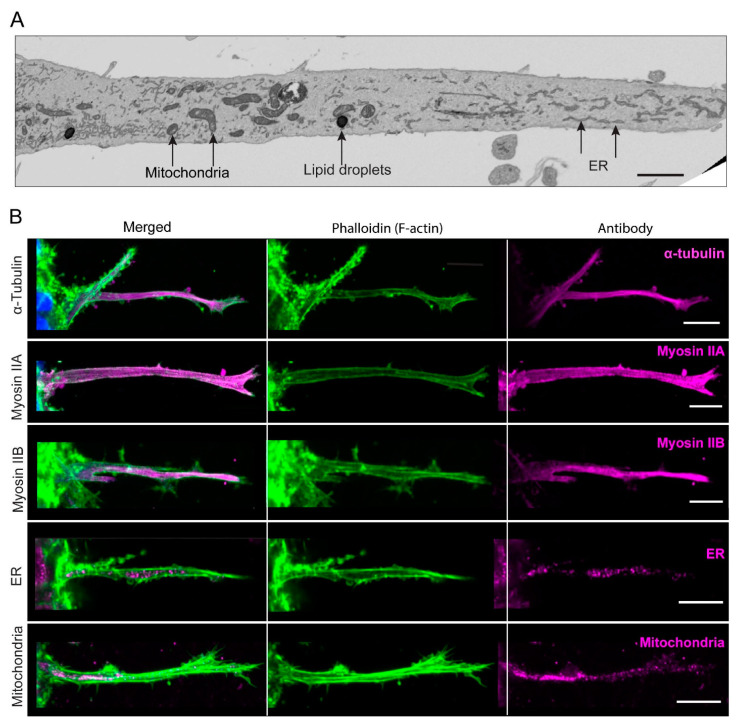
Composition of the very long protrusions extending from spheroids through the BM into collagen matrix. (**A**) Representative image of a long cellular protrusion into a 3D collagen gel by electron microscopy, revealing organelles including endoplasmic reticulum (ER) and mitochondria. (**B**) Representative images of immunofluorescence localization of various organelles and cytoskeletal proteins analyzed in the inhibition studies cited previously or observed by electron-microscopy imaging; green, phalloidin to stain F-actin; magenta, the indicated component; blue, nuclei. Scale bars: (**A**) 2 µm; (**B**) 10 µm.

**Figure 7 cells-12-02474-f007:**
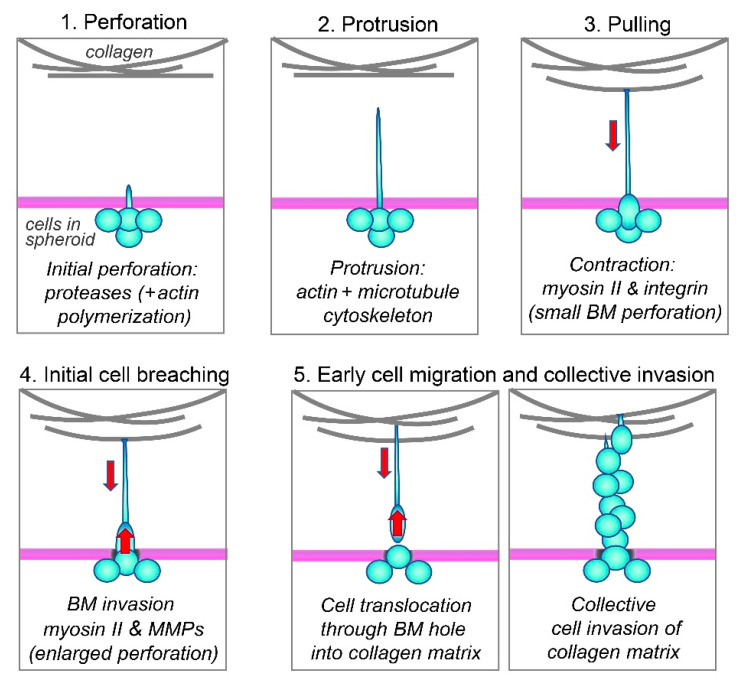
Schematic summary. See text in Discussion for details.

## Data Availability

All the data for this study are available in the published article, and its online Appendix A is deposited and openly available in Figshare, at 10.6084/m9.figshare.24299866. The collagen I, prepared in-house, can be obtained via a materials-transfer agreement (MTA), but it is only available in limited quantities.

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
