# Peer review of "Long Prehensile Protrusions Can Facilitate Cancer Cell Invasion through the Basement Membrane"

_cells, 2023, doi:10.3390/cells12202474_

Round 1
Reviewer 1 Report
In this manuscript from Nazari et al, the Authors describe a mechanism by which some cancer cells breach the basement membrane at first step of the invasion process. This mechanism is based on the extension of long protrusion. Although not completely new, the results are supported by different type of experiments and no concern arise. The manuscript is well written and clear in all sections.
Since the mechanism is not applicable to all cancer cells, as suggested by the Authors, could be a good idea to add in the discussion a section regarding limitation of the study.
Author Response
We have now revised the Discussion with the following statement regarding limitation of the study:
“A limitation of this study is that it is based on only two cancer cell lines. These findings from two cancer cell lines will need to be extended in the future to determine how widespread is this mechanism of protrusion-based BM breaching.”
Reviewer 2 Report
This paper talks about the formation of special unusual long protrusion in human cancer spheroid cells and how it penetrates through the basement membrane. There are several points that needs to be addressed before considering for publication:
Major points
1. In Fig.1A, how does the author know that there’s an increased number of protrusions? It’s only showing one view from part of the image. Please provide a quantification bar graph
2. The authors mentioned about fiduciary inclusions in Fig.1A and B. What is the biological meaning for appearance of this structure in the first 20 hours but then disappears?
3. I couldn’t find the description for 3A and 3B in the manuscript. Please indicate where they are discussed.
4. In Fig.4, are all the inhibitors used here specific to single target protein as indicated? Please consider knockout or knock down of some target genes to validate the results here.
5. It is really interesting to observe mitochondria and ER in the protrusion in Fig.6. Is this special compared to other types of protrusion and what is the potential indication here? I saw this in the Discussion part but maybe consider to go a little further and provide more information.
Minor points
1. In Fig.1B, are all the images coordinating to the time points in Fig.1A? Please specify and label them too.
2. The authors might want to consider moving Fig 2H in front of current Fig.2F-G since they talk about it first.
Author Response
Major points:
- The increasing numbers of protrusions is clearly visible in supplementary video S1, which shows the entire spheroid, and we obtained many other such videos. However, in direct response to the reviewer’s request, we have added a quantification bar graph as the first panel in Figure S1.
- We apologize for the confusion about the fiduciary inclusion. We now explain in the Methods section that “Particulate phase-dense aggregates present in our collagen preparations were found to be useful as fiduciary markers to use in parallel to visualizing the collagen matrix fibers themselves. Because the fibers were so clear, the kymograph analyses were based on collagen fiber translocation, and the fiduciary particles were used only as highly visible markers of collagen movement in a movie montage in Figure 1.” In this particular movie, the particle was drawn to the surface of the spheroid by contractility, and it was no longer visible due to the high refractility of the edge of the spheroid.
- The reviewer is correct that we did not specifically cite Figures 3A and 3B in the manuscript. We now clearly refer to “Fig. 3A and B” when describing the laser ablation experiment.
- The pharmacological inhibitors and inhibitory antibodies used were the standards used in our field, and we had to use such chemical or antibody inhibitors to obtain immediate time-specific inhibition. That is, even though the reviewer is correct that if we had been studying a specific enzyme, a knockout or knockdown approach would be useful. However, doing so for any of the cytoskeletal proteins we examined would not be acceptable to the field because knockout or knockdown of actin, myosin II, tubulin, etc., are known to cause complex phenotypic changes in cell proliferation, morphology, and other properties that would result in problems in interpretation. We therefor now clarify in the Methods section why we used this approach as follows: “We needed to use small molecule and antibody inhibitors rather than knockout or knockdown approaches to be able to apply each treatment acutely in order to avoid long-term cellular changes in gene expression, proliferation, and morphology.”
- This reviewer’s comment/suggestion was useful, and we expanded the Discussion with the following sentence: “Consequently, these long, prehensile cancer cell protrusions appear to function as arm-like extensions of the cell body itself that can both extend and actively contract to pull on extracellular matrix, unlike previously characterized thin anterior protrusive structures devoid of organelles such as mitochondria.” We also added the word “prehensile” to a sentence in Conclusions to underscore their difference from previously described protrusions.
Minor points:
- The reviewer is correct, so to avoid confusion, we now label all the panels in Fig. 1B.
- The reviewer is correct, so we have changed the order of panels so that we now refer to the old Figure 2H as Figure 2F.
Reviewer 3 Report
The work presented to me for review entitled “Long prehensile protrusions may facilitate cancer cell invasion through the basement membrane” is very interesting. The migration of cancer cells contributes to metastasis, which is a very disturbing and undesirable phenomenon. Research on the movement of cancer cells is therefore a very important topic in the field of oncology. The introduction of the article is complete and introduces the reader to issues related to the subject of the work. The research methodology has been described in detail, and the description of the results is fully consistent with that presented in the Figures.
However, I have a few minor comments:
- Primary antibodies should be listed in ‘Materials and methods’
- Please include information that the actin cytoskeleton was visualized by transduction with mNeon Green Lifeact lentivirus, because it is not clear to everyone
- Are the results shown in Fig. 2H are statistically significant? there is no such information in the figure, but there is information in the text 'In addition, blebbistatin treatment resulted in strong concomitant inhibition of outward cancer cell invasion compared to control (91%; Figure 2H, Figure S2)'.
- Please look at Figure 6A again to see if these are definitely lipid droplets because I'm not sure.
Author Response
We appreciate the positive, supportive comments.
- We now list the primary antibodies we used in the Materials and Methods section.
- The mNeon Green Lifeact lentivirus was only used for the live-cell ablation experiments in Fig. 3. We now clarify that the immunofluorescence studies used phalloidin to stain F-actin in Figures 1 and 6.
- Yes, the results in Fig. 2H are highly statistically significant at p < 0.0001 by ANOVA.
- We carefully reconsidered Figure 6A and conclude that based on the specific staining methods we used, the structures are lipid droplets and not other possibilities such as overstained lysosomes or late endosomes; e.g., see https://link.springer.com/article/10.1007/BF00489786